# Measurable Residual Disease (MRD) as a Surrogate Efficacy-Response Biomarker in AML

**DOI:** 10.3390/ijms24043062

**Published:** 2023-02-04

**Authors:** Elisa Meddi, Arianna Savi, Federico Moretti, Flavia Mallegni, Raffaele Palmieri, Giovangiacinto Paterno, Elisa Buzzatti, Maria Ilaria Del Principe, Francesco Buccisano, Adriano Venditti, Luca Maurillo

**Affiliations:** 1Hematology, Department of Biomedicine and Prevention, University of Tor Vergata, 00133 Rome, Italy; 2Hematology, Fondazione Policlinico Tor Vergata, 00133 Rome, Italy

**Keywords:** measurable residual disease, acute myeloid leukemia, molecular biology, surrogate biomarker, precision medicine

## Abstract

In acute myeloid leukemia (AML) many patients experience relapse, despite the achievement of morphological complete remission; therefore, conventional morphologic criteria are currently considered inadequate for assessing the quality of the response after treatment. Quantification of measurable residual disease (MRD) has been established as a strong prognostic marker in AML and patients that test MRD negative have lower relapse rates and better survival than those who test positive. Different techniques, varying in their sensitivity and applicability to patients, are available for the measurement of MRD and their use as a guide for selecting the most optimal post-remission therapy is an area of active investigation. Although still controversial, MRD prognostic value promises to support drug development serving as a surrogate biomarker, potentially useful for accelerating the regulatory approval of new agents. In this review, we will critically examine the methods used to detect MRD and its potential role as a study endpoint.

## 1. Introduction

Acute myeloid leukemia (AML) includes a heterogeneous group of diseases driven by recurrent mutation, chromosomal aberrations and/or altered signaling pathways [1]. Conventional treatment involves two phases: induction and post remissional therapy [the latter including high dose chemotherapy, autologous (AuSCT) or allogeneic stem cell transplantation (ASCT)]. Most patients with AML achieve complete remission (CR) following induction therapy. However, relapse is common and decreases the probability of long-term survival with less than 10% of elderly and 40–50% of younger patients being alive 5 years after the diagnosis. Leukemic relapse arises from residual clonal cells that persist below the threshold of routine measurement by standard morphology whose sensitivity is one in twenty total leukocytes. The identification of these residual leukemic cells (RLC), defined measurable residual disease (MRD), is strongly prognostic for clinical outcome and may have therapeutic implication in the management of AML. 

For over 60 years, cytomorphology has been the standard method for assessing the presence of residual leukemia after treatment. Morphological CR is defined as less than 5% bone marrow blast and no Auer rods in normocellular bone marrow, the absence of extramedullary leukemia, neutrophils greater than 1.000/μL and platelets greater than 100.000/μL [2]. However, this evaluation of response is inadequate to establish the effective residual leukemic burden and the consequent risk of relapse. In this perspective, numerous studies have clearly demonstrated that MRD quantification by other higher sensitivity methods is able to offer a more robust and accurate prognostic tool than morphological criteria [3,4]. In recognition of this, in 2017 the ELN introduced the definition of “complete remission without measurable residual disease” [5]. This reflects the growing evidence that patients who tested MRD negative have, on average, a reduced risk of relapse and better survival than those testing positive [3,6,7].

The aim of this review is to illustrate the current techniques for assessing MRD with their performance and the possible use of MRD measurement as surrogate efficacy-response biomarker to accelerate drugs development/approval in AML. 

## 2. Definition of Biomarkers

The Food and Drug Administration (FDA) defines biomarkers as “a characteristic that is objectively measured and evaluated as an indication of normal biologic or pathological processes, or pharmacologic responses to a therapeutic intervention [8]”. Identification of reliable biomarkers can strongly contribute to the development of new therapies and increase the success rates of clinical trials. Biomarkers can include DNA, proteins, RNA, metabolites measured from biological samples (such as blood, saliva, urine, etc.) and from tissues (i.e., tumor masses). The distinction between patients positive or negative for these markers is usually based on a threshold value applied to quantitative measures and determined through retrospective studies on analyzed biological samples. It is possible to distinguish biomarkers for safety and efficacy. The latter are used to show that all, or a significant proportion, of treated patients have improved. They are classified as prognostic, predictive and pharmacodynamic. Prognostic biomarkers are those that can provide information about the history of the disease, regardless of the treatments that the patient receives. Without a treatment intervention, prognostic biomarkers can predict how a disease will likely progress. It can be used as an enrichment strategy (Enriched Clinical Trials) to select patients who are likely to have peculiar clinical outcomes or progress rapidly. Predictive biomarkers: a biomarker is predictive when it provides information on the activity of specific therapies, allowing patients to be stratified based on their likelihood of responding to a given treatment or not. These markers can also be used in clinical trials as an enrichment strategy to identify a subpopulation that may have a favorable or unfavorable response to a treatment. Pharmacodynamic biomarkers: these biomarkers aid in the determination of a drug’s pharmacological effects and can provide insight into the treatment’s effectiveness. The majority of pharmacodynamic biomarkers are used in phase II trials to better understand how to use a drug and guide dose selection. Finally, more recently, biomarkers have been used as surrogate endpoints in clinical trials if a significant correlation exists between the value of the biomarker and a given clinical endpoint of interest. The advantage of biomarkers is that they can determine the effects of treatment on the clinical endpoint of interest more quickly. Furthermore, in some cases, the direct measurement of clinical responses is not possible; in these cases, biomarkers provide an alternative method for evaluating experimental cures.

## 3. MRD as Emerging Biomarker in AML

In the AML setting the capability to predict outcomes and long-term prognosis is gradually improving as genetic characterization and molecular definitions evolve. As a result, a growing number of molecular and chromosomal biomarkers are being progressively incorporated into AML classification schemes and clinical practice [9]. Some established biomarkers, such as *NPM1* or *TP53* mutations, have well-defined prognostic and predictive significance, whereas others, such as *FLT3* or *IDH1/2* mutations, are the target of new therapies. However, molecular/cytogenetic markers are still insufficient for predicting the individual risk of relapse in AML patients because they do not take into account the extreme biological heterogeneity of leukemic cells as well as the individual patient’s response to treatment. MRD measurement, which summarizes all of these interactions, would be the preferred biomarker for defining patient outcome. Indeed, numerous studies have shown that MRD has prognostic value. Patients in remission who have been tested MRD positive have a higher cumulative incidence rate of relapse and, in many cases, a shorter relapse-free survival (RFS) and/or overall survival (OS) than similarly treated MRD negative patients. The strong correlation between detectable MRD and poor patient outcomes has been confirmed at different timepoints throughout the course of intensive AML therapy, including after one or two courses of induction chemotherapy, after post-remission therapy, and both before and after ASCT [3,10]. Increasing evidence suggests that MRD may also serve as a predictive biomarker in AML for certain treatment scenarios and patient subsets. Since MRD assessments, as stated previously, can stratify patients based on risk of disease recurrence, its measurement can be employed to decide whether a patient should be assigned to a specific type of post-remission therapy. For example, in patients with intermediate-risk AML, an MRD-driven post-remission treatment strategy (ASCT vs. AuSCT, for patients tested MRD positive and negative, respectively) has been shown to improve patient outcome [11]. Similarly, MRD may be useful as a predictive biomarker in determining the optimal conditioning intensity prior to allogeneic HCT [12]. Finally, there is growing evidence that MRD can be used as a biomarker to objectively measure the efficacy of a given drug treatment. Due to the strong relationship between MRD assessments and the risk of relapse and/or survival, there is huge interest in using MRD as a surrogate efficacy-response biomarker (see below). However, to date, limited data from randomized clinical trials have demonstrated that efficacy on the surrogate endpoint (MRD) has a corresponding effect on clinical outcome, i.e., the investigational treatment increases the rate of CR MRD negatives as well as survival when compared with the control treatment. The AMLSG 09-09 trial, which randomized 588 patients with newly diagnosed NPMI-mutated AML to intensive chemotherapy plus ATRA with or without the CD33 antibody-drug conjugate gemtuzumab ozogamicin (GO) is one example. NPM1 transcript levels were significantly lower in the GO arm, and a significantly greater proportion of patients achieved remission without MRD than in the control arm. This was correlated with a lower relapse rate and a better RFS with GO [13]. Similarly, the outcome of patients treated with azacitidine and venetoclax in the randomized phase III trial VIALE-A was evaluated in a recent study. Among 190 patients who achieved composite CR (CR+CRi), 67 (35%) were MRD negative and had a longer duration of response, EFS and OS than MRD positive patients [14]. These findings show that achieving an MRD negative response can predict outcome, even in patients receiving low-intensity combination therapy. Conversely, recent data from the randomized phase 3 QUAZAR AML-001 trial of oral azacitidine (CC-486) vs. placebo maintenance therapy showed that, although the presence or absence of MRD after intensive chemotherapy was a strong prognostic indicator of OS and RFS, however, oral-AZA maintenance therapy provided additional survival benefits when compared with a placebo, regardless of patients’ MRD status at baseline. It is worth noting that nearly 20% of patients with detectable MRD at baseline who were assigned to the placebo arm converted to MRD negativity during follow-up, highlighting the difficulty of using MRD as a potential efficacy-response biomarker in AML [15].

## 4. AML MRD Methods

Different techniques to assess MRD in AML are currently being used and their potential application is under continuous investigation [16,17]. Conventional techniques for MRD analysis include multiparametric flow cytometry (MPFC), reverse transcriptase-quantitative PCR (RT-qPCR)/digital droplet PCR (ddPCR) and next generation sequency (NGS) (Table 1) [16].

These methods are complementary to each other, and the ELN guidelines recommend their combined use when possible [16]. The current recommendations are that patients with acute promyelocyte leukemia (APL), core binding factor (*CBF)* positive AML and AML with *NPM1* mutations should be monitored by molecular methods (RT-qPCR) while all other AML subgroup should undergo MPFC testing [16]. NGS testing has not yet been clinically validated for MRD monitoring. Its use is currently reserved for experimental purposes, often in association with MPFC [4,16]. 

MPFC MRD

The increasing use of MPFC in MRD measurement is due to its wide applicability (>90% of AML), rapidity, specificity and ability to distinguish viable cells from bone marrow debris and dead cells. By using ten-color panels, the current sensitivity limit is put around 10^−3^–10^−5^ [4,18]. The rationale for using MPFC for MRD monitoring relies on the expression on leukemic cells of a combination of antigens and/or flow cytometric physical abnormalities that are absent or very infrequent in normal bone marrow (e.g., cross-lineage expression, over-expression, reduced or absent expression and asynchronous expression). The detection of leukemia-associated immunophenotypes (LAIP) or detection of different-from-normal (DfN) phenotypic patterns represent two complementary strategies of analysis [18,19,20]. The use of LAIP is based on the identification at diagnosis of immunophenotypically aberrant populations (a sort of patient’s “immunologic fingerprint”) that differ from normal hematopoietic cells; these immunological fingerprints are then used to trace residual leukemic cells after treatment. In the latter strategy of analysis, RLC are identified as aberrant cell populations (i.e., LAIPs) within a normal pattern of differentiation by using a fixed antibody panel. Therefore, this strategy of analysis does not require the definition of an immunologic fingerprint at diagnosis. The latter approach can be useful when the immunophenotype is not available at the time of diagnosis but also in cases where the pressure generated by the therapies lead to an “immunophenotypic shifts” defined as the appearance of new phenotypic abnormalities or the disappearance of those previously present [21,22]. Differences between the LAIP and DfN approaches may be minimized if sufficiently large antibody panels (≥8 colors) are used for detection [4,18,23,24]. MPFC-MRD is calculated as the percentage of LAIP positive cells in the total WBC measured in bone marrow. The ELN MRD working party recommend the application of panels including at least eight colors, together with the acquisition of a proper number of events, at least 500.000–1.000.000 cells, excluding debris and CD45 negative events. The threshold of MRD positivity considered reliable for MPFC is set at 0.1%. [18] This threshold guarantees LAIP sensitivity in normal or regenerating BM being above the frequency of any possible background; moreover, it allows for the rapid identification of high-risk patients who require post-remissional intensification with ASCT and /or other innovative treatments. On the other hand, MRD below 0.1% does not exclude the persistence of different amount of RLC and therefore different clinical outcomes. In this view, a threshold of 0.035% was prospectively validated in the context of the GIMEMA AML1310 trial [11], while in another study any pre-transplant MRD positivity (i.e., absence of negativity) was found clinically relevant [25]. It is also possible that, in the future, instead of a “universal” threshold, individual MRD thresholds could be used for different LAIPs based on their different sensitivity and specificity in normal or regenerating bone marrow samples [19]. One of the major concerns with MPFC-MRD is that this technique requires considerable expertise and experience; analysis and data interpretation may have some subjective elements and therefore potential biases, operator-dependent [26]. Approximately 20–25% of MRD-MFC negative patients relapse [27]. Unpredicted relapses may be because of biological and technical reasons. The possible biological explanation may reside in the presence of leukemic subclones that are not detectable with current analysis approaches, and that are chemoresistant, and therefore capable of leading to relapse [27]. Both normal and leukemic stem cells (LSC) reside in the CD34+/CD38- compartment and MPFC is able to distinguish LSC by applying a multicolor analysis, which includes specific biomarkers [28,29]. In the prospective HOVON/SAAK 102 study, the presence of LSC, both at diagnosis or during treatment, compared with MRD represent a negative index, both in the positive MRD population and in the MRD-negative one [30].

From a technical point of view, the role of appropriate sampling plays a crucial role. The poor quality of the bone marrow sample and contamination from peripheral blood are causes of invalidation of the flow cytometric analysis, altering the evaluation of the real value of MRD. Furthermore, the background noise generated by the regenerating marrow, with the presence of myeloid precursors, can results in an unreliable specificity of the method.

b.Molecular Biology-based Approaches

RT-qPCR allows MRD detection in patients with molecular targets that are specific and stable over the treatment course such as *PML-RARα* [31,32], *RUNX1/RUNX1T1* [33], *CBFB-MYH11* [34] and mutated *NPM1* [35,36]. The advantage of using RT-qPCR, a method standardized by the Europe Against Cancer (EAC) consortium, includes high specificity and sensitivity for leukemic cells, high reproducibility between laboratories and reduced risk of contamination [37]. This approach is currently considered the gold standard for patients with AML that harbor the aforementioned mutations. In accordance with the ELN consensus on MRD, the quantitative determination of these genes must be tested at diagnosis, after at least two cycles of therapy and every 3 months for 24 months from the end of treatment [5]. This technique is well established and offers a rapid turnaround time and a superior limit of detection (LOD) to other modalities. The limited applicability (nearly 50% of AML cases) and the possible contamination by non-viable cells carrying the same molecular target is the primary limitation of this approach. 

DdPCR is a newer high-throughput technology that works like conventional PCR, except that it splits DNA samples into thousands of separate reaction chambers. The term ‘droplet’ derives from the method used: a water-oil emulsion that divides the samples into 20.000 droplets. PCR amplification occurs in each droplet, and the wells that contain the target DNA are read as ‘positive’ while the remaining are recorded as negative. DdPCR provides the means to count the absolute number of DNA or RNA molecules in each sample, thus being a technology characterized by high sensitivity and specificity. Unlike RT-qPCR, it is able to amplify target genes, without a standard reference curve. Despite the higher sensitivity compared with traditional RT-qPCR (up to tenfold) and precision, the major pitfall of DdPCR is that for each mutation a specific assay needs to be developed in the same gene [16]. Therefore, the assay is time consuming and costly and the current use is limited to specific recurrent mutations such as *NPM1*, *IDH1* and *IDH2*. 

Next-generation sequencing (NGS) technologies provide the opportunity to study a large number of somatic mutations in one single experiment. The possibility of analyzing many molecular abnormalities at the same time, including cytogenetically normal cases, makes NGS an attractive method for MRD evaluation in AML, especially if one considers that the wide intra-clonal heterogeneity often makes the leukemic clone a moving target [37]. However, although an NGS-based MRD measurement is potentially applicable in all AML patients, several factors that limit its widespread use still need to be evaluated. Indeed, turnaround time, LOD (the sensitivity level is set at about 1%) and the potential for artefacts generation represent the main issues to deal with. Moreover, Clonal Hematopoiesis of Indeterminate/oncogenic Potential (CHIP) can persist in patients in long term clinical remission, and cause misinterpretations of MRD [38,39]. Mutations in genes such as *DNMT3A*, *TET2* and *ASXL1* do not contribute to CHIP and then do not correlate with an increased risk of relapse, again favoring misinterpretations of MRD [40,41]. Finally, NGS technology requires considerable experience, expertise and substantial financial resources. For MRD NGS, there are two methods, both acceptable according to ELN 2022 recommendations: agnostic panel and known mutation (that is specific mutations identified at diagnosis) NGS testing. The former can be more expensive and has the potential to detect new evolutionary genomic signatures. The latter is more specific for detecting any founder mutations; furthermore, this procedure has more robust bioinformatics specifications and may be less expensive. However, according to ELN guidelines, the measurement of MRD using NGS is not yet ready for routine application outside of clinical trials. In the near future, it is likely that a more accurate standardization and validation of the results generated in prospective clinical trials will boost the preferential use of DdPCR and NGS platforms for MRD detection. 

## 5. MRD as a Surrogate Endpoint

Regulatory approval of a drug requires evidence of clinical benefit. In clinical trials for AML, survival estimates such as OS, disease-free survival (DFS) and event-free survival (EFS) are standard endpoints, the meaningful improvement of which is accepted as evidence of a clinical benefit of the experimental drug [42]. When the primary endpoints are survival estimates, these clinical trials may take many years to demonstrate a benefit and demand a huge number of patients “needed-to-treat”, with a delayed access to new drugs and considerable cost burdens. Therefore, one of the key strategies for accelerating patient access to new drugs is the development and validation of surrogate endpoints that can be used to predict a potential survival benefits at an earlier stage. A surrogate endpoint is defined as an alternative endpoint (such as a biological marker, physical sign or precursor event) that can substitute for a direct measure of how patients feel, function, or survive. To serve this purpose, a surrogate endpoint should be strongly associated with the true outcome, lie in the causal pathway for the definitive outcome, manifest early in the course of follow-up and be relatively easy to measure [42,43,44]. Since MRD is a biomarker used in AML patients for prognostic, predictive, monitoring and efficacy-response assessments, it represents the ideal surrogate endpoint of survival benefit. A recent meta-analysis of 81 publications and 11,151 patients, treated front-line with induction and consolidation chemotherapy, has demonstrated a significant association between the levels of MRD and survival, irrespective of age, AML subtype, sample type, time of MRD assessment and MRD detection method. In fact, OS of MRD negative patients doubled the one of those MRD positive (5-year OS 68% vs. 34%; HR 0.36 [95% CI 0.33, 0.39]) [3]. For regulatory approval, even a strong prognostic marker such as MRD cannot be automatically assumed as a surrogate endpoint but should be specifically validated in clinical trials, showing that treatment effects on MRD status correlate to similar changes in OS or DFS. Therefore, in future prospective therapeutic trials for AML, it should be mandatory to include MRD status as a pre-defined endpoint. This implies the need of further efforts to implement the standardization and harmonization of assays across laboratories and the widespread clinical adoption of CR MRD negative as a standard of care of response. 

Current clinical trials using MRD status as a primary endpoint are reported in Table 2. 

## 6. Conclusions

The last few years have been characterized by the introduction of new agents/therapies for AML. However, the duration of clinical trials for providing patients with the access to new molecules is still too long. Although MRD is not officially recognized as a surrogate endpoint, some recent phase I/II trials include CR-MRD as a primary endpoint of their study [45]. Using MRD as a surrogate efficacy-response biomarker to accelerate drug development/approval has already been accepted by regulatory authorities in other hematological malignancies such as multiple myeloma or chronic lymphocytic leukemia and is of great interest as a potential strategy in AML [46]. In fact, conventional cytomorphological response criteria do not allow an adequate assessment of the quality of the response obtained with treatments and therefore are not able to accurately predict the risk of relapse. Beyond light microscopy, new advanced techniques are available to explore the quality of responses and then to estimate MRD, therefore improving patients’ risk-assignment. Implementing international collaborative efforts to compare and harmonize MRD measurement methods is a critical step to strengthen the clinical utility of MRD. Although the use of MRD as a surrogate endpoint requires further and robust validation, the Food and Drug Administration has accepted that “for new drugs that have demonstrated durable CR in patients with relapsed or refractory acute leukemia a bone marrow MRD of less than 0.01% as evidence to support efficacy” [47]. In the era of precision medicine, the inclusion of MRD as a primary endpoint in clinical trials and the MRD-refined evaluation of the efficacy of new agents and combination therapies represents an important step towards in the cure of AML.

## Figures and Tables

**Table 1 ijms-24-03062-t001:** Techniques for MRD analysis.

	MFC	RT-qPCR	NGS
Detects	Immunophenotypically abnormal cell populations	Single molecular abnormality	Multiple molecular abnormalities
Sensitivity	10^−3^ to 10^−5^	10^−4^ to 10^−6^	10^−3^
Advantages	Applicable to >90% of cases; Identifies abnormal stem/progenitor cell compartment; - Easily quantified; - Sensitive; - Quick; -Less expensive than molecular-based approaches - Can assess hemodilution; - Distinguishes between live and dead cells; - Can identify targets for immunotherapy; - High specificity when using LAIP.	- Reproducible; - Highly sensitive; - Can identify therapeutic targets; - Easily quantified and standardized.	- Applicable to >90% of cases; - Can identify therapeutic targets; - Platform can be standardized; - Able to monitor multiple mutations in the same run.
Disadvantages	- Not all AMLs have abnormal immunophenotype (no LAIP-AML); - Phenotype may change over time; - Sensitivity is not uniform between patients; - Best results require fresh material; - Experienced personnel required; - Analysis/data interpretation have subjective elements; - Difficult to standardize; - Variable sensitivity (depending on antibody panel used).	- Not widely applicable; - Genetic abnormalities can persist, even in long-term remission; - Genetic clonal heterogeneity; - Genetic evolution over time; - Emergence or selection of sub-clone(s) at relapse; - Expensive.	- Requires error correction to overcome low sensitivity; - Mutated genes are also present in healthy individuals; - Genetic clonal heterogeneity; - Genetic evolution over time; - Emergence or selection of sub-clone(s) at relapse; - Persistence on some abnormalities even during remission; - Bioinformatic approaches are not uniform; - Expensive.

AML, acute myeloid leukemia; qRT-PCR, quantitative reverse transcription-PCR; MFC, multiparameter flow cytometry; NGS, next-generation sequencing; PCR, polymerase chain reaction.

**Table 2 ijms-24-03062-t002:** Current Trials involving MRD as a primary endpoint.

Trial Number	Phase	N	Age	Group	Treatment	Primary Outcome	Technique
NCT04209712	Early Phase I	6	1 to 80	MRD positive after chemotherapy and no SCT	NK infusion with consolidation therapy	MRD	MPFC
NCT04196010	I	13	>18	R/R AML or other high-grade myeloid neoplasms	CI-CLAM	CR_MRD_	Not specified
NCT03701295	I/II	1	>18	AML with 11q23 mutation	Pinometostat, Azacitidine	CR_MRD_	Not specified
NCT04000698	I/II	25	>25	Pediatric R/R AML and ALL	Different targeted therapies	MRD negativity at different time points	Not specified
NCT03699384	I/II	0	>18	MRD positive AML	Azacitidine Avelumab	Sustained MRD negativity	MPFC
NCT02614560	I/II	14	18–75	R/R AML	Vadastuximab Talirine	Rate of MRD negativity	Not specified
NCT04347616	I/II	23	>18	R/R AML	NK cell therapy	MRD levels	MPFC, PCR
NCT03021395	I/II	300	14–55	AML after consolidation	Decitabine	MRD clearance rate	Not specified
NCT04086264	I/II	274	18–120	CD123^+^ AML	IMGN632, Venetoclax, Azacitidine	MRD levels	MPFC
NCT04284787	II	76	>60	Unfit AML	Pembrolizumab, Azacitidine, Venetoclax	CR_MRD_	Duplex sequencing, MPFC
NCT04214249	II	124	>18	AML	Pembrolizumab plus intensive chemotherapy	CR_MRD_	MPFC
NCT03737955	II	36	>2	MRD positive AML	GO	MRD clearance rate	MPFC, PCR
NCT03150004	II	90	>18	R/R secondary AML	CLAG-M	CR_MRD_	MPFC
NCT04476199	II	100	60–75	De novo AML and bridge to alloSCT	Venetoclax, Decitabine	CR_MRD_	MPFC, Cytogenetics, qRT-PCR
NCT03573024	II	36	18–59	De novo AML	Venetoclax, Azacitidine	CR_MRD_	MPFC
NCT03654703	II	100	3–18	Pediatric R/R AML	Cyclophosphamide regimes	CR_MRD_	MPFC
NCT01677949	II	0	<60	ALL and AML	Clofarabine, Cyclophosphamide, Etoposide	MRD conversion	MPFC, PCR
NCT03697707	II	20	>18	R/R AML with persistent MRD positivity	Dendritic cell theraphy	MRD conversion	MPFC
NCT00863434	II	2	18–75	MRD positive AML	Clofarabine, Cytarabine	MRD conversion	MPFC
NCT00965224	II	50	>18	AML and MM	Dendritic cell therapy	MRD negativity/ Manteinance	WT1-PCR
NCT03665480	II/III	122	14–65	De novo AML	G-CSF	MRD level	Not specified
NCT04093505	III	28	>60	De novo and post-remission AML	GO, Glasdegib	MRD negativity	MPFC
NCT04168502	III	414	18–60	De novo AML, favorable and intermediate risk	Gemtuzumab, Glasdegib	MRD negativity	Not specified
NCT01828489	III	300	0–80		Cytarabine/ Fludarabine, DaunoXome, Etoposide/ Cytarabine	MRD levels	MPFC
NCT01347996	IV	84	>18	AML patient in first CR	Histamine, IL-2	MRD	qRT-PCR
NCT03549351	Observational	51	Children, Adults, Older adults	AML	/	Correlation between MRD and Survival	MPFC

N, number of patients; CI-CLAM, continuous infusion chemotheraphy = cladribine, cytarabine, mitoxantrone; CR MRD, complete remission measurable residual disease; MPFC, multiparameter flow cytometry; PCR, polymerase chain reaction; GO, Gentuzumab Ozogamicin; R/R, relapse/refractory; CLAG-M, cladribine, cytarabine, mitoxantrone and filgrastim; qRT-PCR, quantitative real time polymerase chain reaction; MM, multiple myeloma; G-CSF, G colony stimulating factor.

## Data Availability

Not applicable.

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
