# Peer review of "Measurable Residual Disease (MRD) as a Surrogate Efficacy-Response Biomarker in AML"

_ijms, 2023, doi:10.3390/ijms24043062_

Round 1
Reviewer 1 Report
The topic is interesting and a summary of the role of MRD in AML will definitely contribute to the field of the predictive biomarkers of the outcome of AML. The authors compared the techniques for measuring MRD in AML, reviewed the published studies and current trials involving MRD as a primary endpoint systematically, and put up with that MRD as a surrogate efficacy-response biomarker in AML.
One minor comment: the name of the gene should be in italic, such as NPM1, FLT3.
Author Response
One minor comment: the name of the gene should be in italic, such as NPM1, FLT3.
Response: all gene names mentioned in the manuscript have been written in italic (please see the attachment).
Reviewer 2 Report
1. Gene names should be all italicized through entire paper.
2. Please correct NPM name to NPM1 gene (Line: 87).
3. For MRD NGS there are two methods, which both are acceptable by ELN 2022 recommendations: 1. Agnostic NGS 2. Known mutation NGS testing. The Agnostic can be more expensive and have potential to detect the new evolutionary genomic signature. The Known mutations is more specific for detect any founder mutations and methods have more robust bioinformatics specify and can be less expensive.
Author Response
1. Gene names should be all italicized through entire paper.
Response 1: all gene names reported in the manuscript have been italicized
2. Please correct NPM name to NPM1 gene (Line: 87).
Response 2: NPM name has been corrected to NPM1 gene
3. For MRD NGS there are two methods, which both are acceptable by ELN 2022 recommendations: 1. Agnostic NGS 2. Known mutation NGS testing. The Agnostic can be more expensive and have potential to detect the new evolutionary genomic signature. The Known mutations is more specific for detect any founder mutations and methods have more robust bioinformatics specify and can be less expensive.
Response 3: the suggested paragraph was included in the manuscript (please see the manuscript rev1)